# Determinants of Adherence in Time-Restricted Feeding in Older Adults: Lessons from a Pilot Study

**DOI:** 10.3390/nu12030874

**Published:** 2020-03-24

**Authors:** Stephanie A. Lee, Caroline Sypniewski, Benjamin A. Bensadon, Christian McLaren, William T. Donahoo, Kimberly T. Sibille, Stephen Anton

**Affiliations:** 1Department of Aging and Geriatric Research, Institute on Aging, University of Florida, Gainesville, FL 32610, USA; csypniewski@ufl.edu (C.S.); bensadon@ufl.edu (B.A.B.); mclaren@ufl.edu (C.M.); ksibille@ufl.edu (K.T.S.); 2Department of Medicine, University of Florida, Gainesville, FL 32610, USA; troy.donahoo@medicine.ufl.edu; 3Department of Clinical and Health Psychology, University of Florida, Gainesville, FL 32610, USA

**Keywords:** weight loss, intermittent fasting, fat loss, sarcopenia, body composition

## Abstract

Time-restricted feeding (TRF) is a type of intermittent fasting in which no calories are commonly consumed for approximately 12–18 hours on a daily basis. The health benefits of this eating pattern have been shown in overweight adults, with improvements in cardiometabolic risk factors as well as the preservation of lean mass during weight loss. Although TRF has been well studied in younger and middle-aged adults, few studies have evaluated the effects of TRF in older adults. Thus, the goal of this study was to evaluate older-adult perspectives regarding the real-world advantages, disadvantages, and challenges to adopting a TRF eating pattern among participants aged 65 and over. A four-week single-arm pre- and post-test design was used for this clinical pilot trial TRF intervention study. Participants were instructed to fast for approximately 16 h per day with the daily target range between 14 and 18 h. Participants were provided with the TRF protocol at a baseline visit, along with a pictorial guide that depicted food items and beverages that were allowed and not allowed during fasting windows to reinforce that calorie-containing items were to be avoided. The trial interventionist called each participant weekly to promote adherence, review the protocol, monitor for adverse events, and provide support and guidance for any challenges faced during the intervention. Participants were instructed to complete daily eating time logs by recording the times at which they first consumed calories and when they stopped consuming calories. At the end of the intervention, participants completed an exit interview and a study-specific Diet Satisfaction Survey (Table 1) to assess their satisfaction, feasibility, and overall experience with the study intervention. Of the 10 participants who commenced the study (mean age = 77.1 y; 6 women, 4 men), nine completed the entire protocol. Seven of the ten participants reported easy adjustment to a 16-hour fast and rated the difference from normal eating patterns as minimal. Eight participants reported no decrease in energy during fasting periods, with greater self-reported activity levels in yardwork and light exercise. Adverse events were rare, and included transient headaches, which dissipated with increased water intake, and dizziness in one participant, which subsided with a small snack. The findings of the current trial suggest that TRF is an eating approach that is well tolerated by most older adults. Six participants, however, did not fully understand the requirements of the fasting regimen, despite being provided with specific instructions and a pictorial guide at a baseline visit. This suggests that more instruction and/or participant contact is needed in the early stages of a TRF intervention to promote adherence.

## 1. Introduction

Aging is often associated with a host of biological changes that contribute to a progressive decline in cognitive and physical function, frequently leading to a loss of independence and increased risk of mortality. The life expectancy of adults in many industrialized countries continues to increase [1], with persons aged ≥65 years representing the fastest growing segment of the US population [2]. While prolongation of life remains an important public health goal, of even greater significance is the extension of healthspan, often defined as continued intact functional capacity, and delay of the physiological changes that result in disease and disability [3,4]. For these reasons, there is a longstanding interest in understanding the biopsychosocial and functional determinants of successful aging [5]. Although many factors can contribute to functional decline, loss of muscle mass (sarcopenia) in particular, has been consistently linked to functional decline during aging [6].

Globally, sarcopenia has become a major health challenge, and is now recognized as a medical condition across the world [7,8]. In a recent multi-ethnic study (MEMOSA—Multi-Ethnic Molecular determinants of Sarcopenia) involving participants from Singapore, the UK, and Jamaica [9], the genome-wide transcriptomic profiles of skeletal muscle biopsies in 119 older men diagnosed with sarcopenia compared with age-matched controls were examined using high-coverage RNA sequencing. The novel and important finding of this study was that mitochondrial bioenergetic dysfunction was the strongest molecular signature of sarcopenia in men irrespective of ethnicity. Specifically, sarcopenia was associated with major impairments of oxidative phosphorylation, mitochondrial dynamics, and mitochondrial quality. Such findings strongly suggest links between mitochondrial health, muscle quality, and physical function in older adults. 

Exercise is widely known to improve mitochondrial health, potentially by providing a hormetic challenge that induces mitochondrial biogenesis [10]. Another potential intervention strategy for enhancing the metabolic flexibility of the mitochondria is intermittent fasting (IF), or more specifically, time-restricted feeding (TRF) [11]. This type of eating pattern involves a cessation in caloric intake commonly for 12–18 hours daily and has been shown to be sufficient to induce the metabolic switch from glucose to ketones as a source of energy for the mitochondria [12,13,14]. Specifically, this shift in metabolism takes place when nutrient availability is low and occurs at the point of negative energy balance when liver glycogen stores are depleted and fatty acids are mobilized (typically beyond 12 h after cessation of caloric intake) [15,16].

In contrast to traditional caloric restriction paradigms, food is not consumed during designated fasting time periods but is typically not restricted during designated eating time periods. The length of the fasting time period can also vary but is frequently 12 or more continuous hours. There are many types of intermittent fasting approaches, but the two most popular and well-studied approaches are alternate day fasting (ADF) or alternate day modified fasting (ADMF) and TRF. Alternate day or alternate day modified fasting involves consuming no or very little food on fasting days and then alternating with a day of unrestricted food intake or a “feast” day. Time-restricted feeding interventions differ from ADF interventions in that individuals engage in daily fasts between 14 and 18 hours. Findings from a recent review indicate participants generally have high levels of adherence (range = 77% to 98%) with no serious adverse events to fasting regimens ranging in duration from two weeks to one year [17].

Several clinical trials now indicate that a TRF eating pattern can reduce fat mass with retention of lean mass in younger and middle-aged adults [18,19,20,21]; however, the effects of TRF are not well understood in adults aged 65 and older. We recently reported that a four-week TRF eating pattern was sufficient to induce weight loss and produce small but clinically meaningful improvements in physical function in overweight older adults [22]. Although these findings are promising, they are limited by the short duration of the intervention. Therefore, the present study aimed to better understand factors affecting adherence and feasibility of TRF in an older adult population. 

## 2. Methods

As described previously [22], ten overweight, sedentary, older adults aged 65 and older with mild to moderate functional limitations were recruited to participate in the *Time to Eat* pilot study. All participants were living independently in the community. Primary outcomes were feasibility, tolerability, and safety in overweight, sedentary older adults over four weeks using a single-arm pre–post design. Secondary outcomes included changes in body weight, waist circumference, cognitive and physical function, health-related quality of life, and adverse events [22]. The overarching goal of the present study was to evaluate participant perspectives who were enrolled in the *Time to Eat* pilot study regarding the real-world advantages, disadvantages, and challenges to adopting a TRF eating pattern via weekly phone interviews, a diet satisfaction survey, and an exit interview during the study.

### 2.1. Intervention

A four-week TRF single-arm clinical pilot trial was employed to test the study objectives. All participants were asked to abstain from any caloric intake during the targeted fasting window of 16 continuous hours and consume ad libitum during the eating window. Fasting times were chosen by the participant and were allowed to vary each day. The first week involved a gradual increase to a full 16-hour fasting period (Days 1–3 fast for 12–14 hours per day, Days 4–6 fast for 14–16 hours per day, Days 7–28 fast for 16 hours per day). Participants were encouraged to hydrate during fasting times. There were no dietary restrictions on the amount or types of food consumed during the 8-h feeding window, and participants were allowed to choose a time frame that best fit their lifestyle. 

To promote adherence to the intervention, participants were provided with directions on how to follow the TRF protocol at the baseline visit. During this visit, participants received a pictorial flyer for future reference which depicted food items that were allowed and not allowed during fasting windows. On the flyer, “Go Foods” were written in green (water, diet soda, unsweetened teas, sugar free gum, black coffee) and “No Foods” were written in red (anything with calories, including coffee creamer, sweet teas, alcohol, snacks, drinks with calories). Participants also received an eating time log, which they were instructed to complete by recording the times when they first consumed calories and when they stopped consuming calories each day. 

The trial interventionist called each participant weekly to promote adherence, review the protocol, monitor for adverse events, and provide support and guidance for any challenges faced during the intervention. A semi-structured, open-ended interview guide was used for the calls to inquire about daily activities, changes to normal routine, and any changes in health. During the call, the eating time log was reviewed, and support and guidance were provided for any challenges faced due to the intervention. An intervention-specific interview guide was used to routinely monitor adherence and adverse events, and to obtain direct participant feedback at multiple time points throughout the trial. The same questions were asked during each call, and calls concluded with open discussion. Participant answers and comments were documented during the phone calls. The questions asked during each phone call are displayed in Appendix A.

### 2.2. Adherence

Self-reported adherence to the study intervention was measured using the eating time log. Participants were considered adherent if they fasted between 14–18 hours per day during weeks two through four of the intervention. For this intervention, participants were allowed to pick the times of day in which they fasted as well as their eating window. 

### 2.3. Outcome Measures

Diet Satisfaction Survey. At the end of the intervention, participants completed a study-specific Diet Satisfaction Survey (Table 1) to assess their satisfaction, feasibility, and overall experience with the study intervention. The Diet Satisfaction Survey contained 22 items and categorically measured respondents’ attitudes by asking the extent to which they agreed or disagreed with a particular statement on a 5-point Likert scale (strongly disagree, disagree, neutral, agree, and strongly agree). The scores were summed and questions were grouped into one of three domains: biological, psychological, and socio-environmental.

Exit Interview. Exit interviews were also conducted in person by the study interventionist during the follow-up assessment visits. Interview format mirrored that of the weekly phone calls and included additional questions measuring interest and the likelihood of continuing the TRF eating pattern. The interview concluded by allowing the participant to offer future suggestions and modifications if desired. The Specific Exit Interview Questions asked are displayed in Appendix A. 

### 2.4. Analyses

Using the information obtained during the weekly phone calls and exit interviews, participant views on the perceived advantages, disadvantages, and challenges to adopting the TRF regimen were explored through qualitative analyses to better understand how older adults can successfully adopt TRF. A constant-comparison coding process was used to categorize and compare interview data for analysis purposes [23]. A trained researcher (SL) open-coded each interview and mapped data within the three primary constructs of the biopsychosocial model (biological, psychological, socio-environmental) to systematically categorize these three primary factors in their interactions in the TRF intervention. Open coding was completed by hand instead of using data mining software in order to take on the full context of the interviews. Positive and negative categories within each of the three primary themes were documented.

## 3. Results

Nine of the ten participants who commenced the study (mean age = 77.1 y; 6 women, 3 men) completed the entire protocol. One participant was considered non-adherent as the participant did not complete phone calls during the intervention or post-intervention assessments. However, this participant did complete an exit interview but not the Diet Satisfaction Survey post intervention. Thus, exit interviews were completed with all 10 participants who were enrolled in this study. Four participants completed all three weekly phone calls with the interventionist, three participants completed two calls, one participant completed one call near study end due to the wrong phone number being provided, and one participant did not complete any phone calls due to being on vacation. 

Self-reported mean adherence to the TRF regimen was 84%, measured by daily eating time logs [22]. Few adverse events were reported during this intervention. Specifically, two participants experienced headaches during fasting periods, which resolved following an increase in water intake. One participant experienced dizziness, which resolved after having a small snack.

Over the four-week TRF intervention, the average reported start time for the participant eating period was 10:21 AM (range = 6:56 AM–1:25 PM) and the average reported stop time was 6:38 PM (range = 5:08 PM–9:00 PM), respectively. During week 1, the average reported start time of the first meal was 9:23 AM. During weeks 2 to 4, the start time of the eating period shifted later in the morning by a little over an hour, with participants reporting an average start time of 10:33 AM during week 4. The average eating stop time occurred 27 minutes earlier over the four-week study, concluding at 6:49 PM during week 1 and at 6:22 PM during week 4. The self-selected start and stop times for each participant’s eating window are displayed in Figure 1. 

Participant answers to the questions on the Diet Satisfaction Survey revealed specific adherence-related barriers and facilitators within each of the three primary domains (biological, psychological, social) of the biopsychosocial model. Summed scores and percentages are shown in Table 1. 

### 3.1. Biological Factors

Five participants reported easy adjustment to a 16-h fast, agreeing that fasting got easier as the study went on. Seven participants reported that it was not difficult to eat enough calories within the 8-h window. After the first few days of fasting, only one participant reported uncomfortable hunger during the study, but this participant misunderstood the protocol and was only eating one meal a day. Six of the nine participants stated that overeating was not problematic during the eating window and two participants neither agreed nor disagreed. Normal energy levels were retained throughout the intervention, with eight of the nine participants reporting “disagree” when asked if energy levels decreased while fasting. Only one participant indicated that fasting interfered with normal sleep patterns. Participant responses to the five questions within the biological domain of the Diet Satisfaction Survey are displayed in Appendix A. 

### 3.2. Psychological Factors

Subjective mood and quality of life were unaffected in six of the nine participants, and seven participants expressed eagerness to participate in a similar study again. Despite this, participant comprehension of the TRF protocol was lower than expected, with many participants not fully understanding instructions regarding calorie consumption during fasting times. For example, during weekly phone calls, some participants stated they ate snacks in the evening before bed but did not record this in their eating window. Additionally, low-calorie foods were often confused with no-calorie foods, and not recorded even after the interventionist reiterated that any food item with calories must be documented. One participant misunderstood the regimen and only consumed one meal per day throughout the study, which was not revealed until the exit interview. Additionally, five participants thought that TRF was being examined solely for weight loss despite the interventionist explaining that other health outcomes were being evaluated. Due to this misunderstanding, two participants focused on consuming small or low-calorie foods during the eating window rather than eating as normal.

Six of nine participants agreed that they could adhere to the TRF eating pattern for six months, however, this number gradually decreased when asked about the feasibility of maintaining a TRF eating pattern for 12 or 24 months. Six of the participants agreed that fasting became easier over the study period, while two disagreed and one neither agreed nor disagreed. Five of the nine participants stated that they would continue fasting after concluding the study, while two had no preference and two indicated they would not continue. Participant responses to the ten questions within the psychological domain of the Diet Satisfaction Survey are displayed in Appendix A.

### 3.3. Socio-Environmental Factors

Family support was central to participant adherence. Although not formally enrolled in the trial, the spouses of participants often altered their eating patterns to accompany their partner. Seven participants were able to space out their meals during their chosen eight-hour eating window rather than consuming all daily calories in one sitting, with four participants reporting inconvenience in eating all of their meals during the eating window. Environmental challenges to TRF adherence included social events during which food was served outside the participant’s eating window, as well as changes in work schedules that interfered with the timing of meals. Regular doctor appointments for the participant or their spouse requiring long commutes also extended fasting times beyond the 14-18 hour goal for a few individuals. Additionally, several participants reported that they would not want to follow this eating pattern during holidays or vacations. Notwithstanding these challenges, only one participant stated they would not recommend TRF to a friend, and five believed people would follow this eating pattern if recommended by their healthcare provider. Participant responses to the seven questions within the socio-environmental domain of the Diet Satisfaction Survey are displayed in Appendix A.

## 4. Discussion

The findings of the current trial suggest that TRF is an eating approach that is well tolerated by most older adults. However, six participants did not fully understand the requirements of the fasting regimen, despite being provided with specific instructions and a pictorial guide at a baseline visit. Among these six participants, three reported consuming snacks during fasting periods, two participants confused low-calorie with no-calorie items, and one participant thought they were only allowed to eat one meal a day. Thus, these findings suggest that more instruction and/or participant contact is needed in the early stages of a TRF intervention to promote adherence. 

Most participants reported limited physical discomfort caused by this eating pattern. After the first few days of fasting, only one participant reported discomfort related to hunger, and this was likely due to misunderstanding the eating pattern directions. The few reported side effects included transient headaches which dissipated with increased water intake, and dizziness in one participant which subsided with a small snack. 

While it is challenging to separately measure biological, psychological, and socio-environmental factors related to eating, energy, and satiety, our findings suggest each domain is relevant to TRF adherence. From a biological perspective, all but one participant perceived they were consuming the same or higher amount of calories as they did prior to beginning the intervention. It is also noteworthy that eight out of nine participants disagreed that fasting decreased energy levels, with greater self-reported activity levels in both yardwork and light exercise. Eight participants indicated on the Diet Satisfaction Survey that the TRF intervention did not negatively affect their sleep, with only one participant reporting that fasting interfered with their normal sleep patterns. 

From a psychological perspective, most participants also expressed positive attitudes on the phone calls throughout the study and seven of the nine participants reported feeling eager, motivated and excited to continue with the intervention. Many had heard of IF regimens and wanted to experience this eating pattern. Despite this, participant comprehension of the TRF protocol was lower than anticipated, with six participants not fully understanding instructions regarding avoiding calorie consumption during fasting times. Two participants also had difficulty differentiating foods and beverages with low versus no calories. Additionally, participants were initially inaccurate and inconsistent when reporting their food intake times. To maximize protocol comprehension, it is recommended that future interventions provide even more frequent contact (e.g., bi-weekly) during the initial intervention period to ensure the participant understands the protocol. Similar to calorie restriction interventions, ongoing contact is advisable and can equip participants with adherence promoting behavioral modification techniques and strategies, and continued monitoring for adverse events. 

Socio-environmental factors also served as both barriers and facilitators to adherence. Consistent with prior literature [24], participants were often positively influenced by their partners. Many participants reported receiving significant support from spouses, some of whom even changed their eating patterns to be in synchrony with them during the intervention. These reports were highly encouraging, as successful behavior modification requires disrupting the socio-environmental factors that cue habitual behavior [25], and spousal eating times represent an important factor that could cue eating in the participants [26]. On the other hand, a few socio-environmental factors emerged as barriers to adherence. Also consistent with previous studies [27,28,29], pressures from work, long commutes, vacations, and social engagements represented barriers to adherence to intervention for some participants. 

Despite strong evidence indicating that lifestyle intervention programs involving diet, exercise, and behavior modification can reduce risk factors for many chronic diseases and improve physical function, long-term adherence to lifestyle interventions to date is notoriously low [30]. Consequently, the “adherence problem” represents an important challenge to weight loss interventions [30]. Findings from a recent review of 27 studies indicate that participants generally have high levels of adherence (range = 77% to 98%) to different types of fasting regimens, including ADF and TRF [17]. Thus, future trials are needed to evaluate the potential that this eating pattern may have for enhancing long-term weight loss. 

There were a few notable strengths of this study. First, participants received personalized attention throughout the intervention, which allowed for discussion of individual challenges and tailored solutions to help participants adopt this new eating pattern. This contact was provided through weekly phone calls to check on their adherence and assist them in problem-solving any challenges they were experiencing following the intervention. Second, adherence was carefully tracked throughout the study, as participants reported the time of their first and last meal in an eating time log each day. The participants then reported their daily start and stop times during weekly phone calls and adjustments were discussed if needed. Third, adverse events were assessed, and potential solutions were offered during the calls. We also conducted exit interviews to obtain each participant’s perspective on what challenges and what changes, if any, they would recommend the intervention in future trials. 

There were also several limitations to this study, including difficulty in contacting some participants via telephone during the intervention. Additionally, only one coder was used for analysis, and triangulation was not used to cross-check data collection. In future studies, a second coder will be used to verify the coding scheme. As this was a pre–post pilot study of short duration, a small sample size was used, which limits the generalizability of the results as well as our ability to make conclusive statements. Response bias is also a possible limitation of this study, as participants may have only reported what they thought the interventionist wanted to hear in the interviews, or second-guessed what the interventionist was asking and altered their answers. Individual interpretation of the questions asked on the Diet Satisfaction Survey may have also varied. Future studies should utilize larger sample sizes to ensure sufficient power to detect both pre–post differences and between-group differences. 

## 5. Conclusions 

Most (6 of 10) participants appreciated the simplicity of the time-restricted eating pattern and reported willingness to continue this eating pattern with slight modification such as decreased daily fasting times or a weekly “unrestricted feeding” day. Comprehension of the fasting regimen and guidelines varied, indicating more participant contact and/or education is needed during the initial stages of the intervention. Additionally, baseline group instruction about the protocol with weekly in-person group sessions may enhance understanding and social support. Motivation to follow the protocol was strongly guided by weight loss, as many participants thought this was the goal of the study. Interventions with older adults should consider these factors to optimally support lifestyle modification and protocol adherence.

### Clinical Implications

This study provides exciting preliminary data on the potential beneficial effects and feasibility of TRF in older adults with very few risks associated with the intervention. In addition to the previously reported weight loss outcome and high adherence to the intervention [22], most participants adjusted relatively easily to the fasting window of 14–18 hours. Adverse effects were rare and were quickly remedied. No decrease in energy during the fasting time was reported by eight participants. There was also a carry-over effect, with some family members participating and a high likelihood the participants would recommend this intervention to a friend. While these findings are promising, more work needs to be done before a TRF intervention can be effectively implemented into clinical practice. The primary challenge relates to the participants’ initial misunderstanding of what they were and were not allowed to consume during the 14–18 hour fasting period. The addition of dietary monitoring in which participants record their food intake, as well as the start and stop times, may be one approach to enhance adherence levels. This approach has proven essential for most behavioral weight loss programs, so it may be beneficial to helping older adults understand the key distinction between the fasting and non-fasting time periods. Other common adherence barriers expressed by participants included program adherence during social events, changes in work schedule, or vacations. Nevertheless, there is more opportunity to overcome these barriers due to the flexibility of TRF and the ultimate ability to focus on a yes or no behavior (eating vs. fasting) rather than consciously eating less or counting calories with traditional diets.

## Figures and Tables

**Figure 1 nutrients-12-00874-f001:**
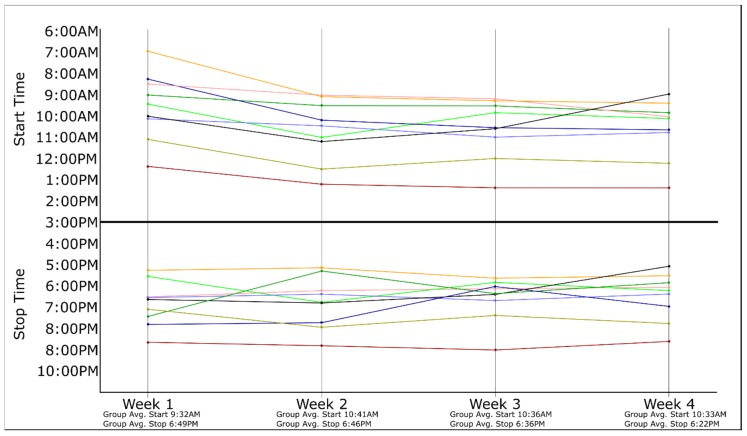
Self-selected start and stop times for each participant’s eating window. N = 9 for each of the four weeks. Each participant’s average weekly self-reported start/stop times are indicated by differing colors, with each line representing a single participant. The time between “Start Time” and “Stop Time” is indicative of each participant’s eating window.

**Table 1 nutrients-12-00874-t001:** Participant responses to the Diet Satisfaction Survey.

Questions	Strongly Disagree (%)	Disagree (%)	Neutral (%)	Agree (%)	Strongly Agree (%)
**Biological**					
It was difficult to eat enough calories within the 8 hour window.	3(33)	4(44)	1(11)	1(11)	0(0)
I would eat more than normal amount of food during feeding period.	1(11)	5(56)	2(22)	1(11)	0(0)
I was uncomfortably hungry while fasting	0(0)	7(78)	1(11)	0(0)	1(11)
I had less energy than usual while fasting.	3(33)	5(56)	0(0)	0(0)	1(11)
I had difficulty falling asleep, staying asleep, or waking during fasting.	2(22)	3(33)	3(33)	1(11)	0(0)
**Psychological**					
Fasting became more difficult as study went on.	4(44)	2(22)	1(11)	1(11)	1(11)
Fasting became easier as the study went on.	0(0)	2(22)	1(11)	3(33)	3(33)
My mood improved while intermittent fasting.	0(0)	2(22)	6(67)	0(0)	1(11)
My mood worsened while intermittent fasting.	2(22)	1(11)	5(56)	1(11)	0(0)
Fasting was more difficult than others previously tried.	2(22)	3(33)	3(33)	0(0)	1(11)
My overall quality of life improved during the IF diet.	0(0)	2(22)	6(67)	1(11)	0(0)
I could continue to IF for 6 months.	1(11)	2(22)	0(0)	6(67)	0(0)
I could continue to IF for 12 months.	2(22)	2(22)	1(11)	4(44)	0(0)
I could continue to IF for 24 months.	3(33)	1(11)	3(33)	2(22)	0(0)
I plan to continue IF after this study.	0(0)	2(22)	2(22)	5(56)	0(0)
I would participate in this study again.	1(11)	1(11)	1(11)	5(56)	1(11)
**Socio-environmental**					
I consumed all of my calories in one sitting.	3(33)	4(44)	1(11)	1(11)	0(0)
Fasting periods made tasks and work and home more difficult.	2(22)	3(33)	2(22)	1(11)	1(11)
Fasting periods negatively impacted my Social life.	2(22)	3(33)	2(22)	2(22)	0(0)
It was inconvenient to restrict my food intake to an 8-hour time frame.	0(0)	4(44)	1(11)	3(33)	1(11)
If recommended by their doctor, I think people would follow the IF diet.	0(0)	0(0)	4(44)	5(56)	0(0)
I would recommend this study to a friend.	1(11)	0(0)	2(22)	4(44)	2(22)

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
