# Peer review of "Determinants of Adherence in Time-Restricted Feeding in Older Adults: Lessons from a Pilot Study"

_nutrients, 2020, doi:10.3390/nu12030874_

Round 1
Reviewer 1 Report
I am happy to see this pilot pre-post TRF study done in an older population as a first step in evaluating the feasibility, tolerability and safety of the intervention regimen in this population. The authors are to be congratulated on their consideration of these factors and population. The findings are promising. Some concerns dampen my enthusiasm of the manuscript in its present form.
MAJOR CONCERNS.
Abstract.
Line 12. Slight mis-characterization of Delete “few or”. TRF design is that no calories are to be consumed during the fasting period. Further, TRF is not a prescribed fasting length of 12-14 hrs, there are shorter and longer regimens that have been published. Best to describe simply that 12-14 hrs is “often” or “commonly” used in study designs. Here in abstract, also Line 74.
Study design should be specifically mentioned in abstract: Pilot, one-arm pre-post intervention study.
Line 26. Demographic data should be for intention-to-great cohort (N=10). Unclear, whether data show is for the 10 enrolled or the 9 who completed. Please address and add BMI metrics.
How and how often data about feasibility, acceptability, and other input on intervention outcomes was collected should be reported in abstract, as these are main outcomes.
Outcomes should be quantitatively (not using percentiles) in abstract. Terms like “majority”, “some”, and “rare” are too vague.
The “Methods” section of the abstract should contain the intervention length and the design (daily 16hr fast)
Challenges in the participants’ understanding of the protocol is an important finding. Strongly suggest adding here, or it will be missed by many who would find it very useful to know.
Methods.
Line 87. Please clarify. Is this manuscript a report of secondary outcomes analyses from the same research study presented in reference #19? If so, this must be made completely clear that this the same study and participants and not simply the same protocol. It is the same study, are there any data presented here (text, figures, tables) that were presented in ref #19?
Line 87. Define “mild to moderate functional limitations” in this manuscript. Where were the participants living? Please provide information on this. Free-living or living in senior centers with common dining? These factors are important for this population as well as the intervention feasibility.
Line 91. List of secondary outcomes sounds like ref #19. No reference provided here. Please add. Why were the “secondary outcomes” of the study reported first? Seems that this manuscript is reporting secondary outcomes of the overall study. Please explain.
The methods section should include a CONSORT-type diagram. There is a pilot study version of this. What was recruitment efficacy like, how many individuals screened prior to enrollment, why were people excluded and screening out, etc.?
Results.
Line 151. It is standard to report demographics of the intention-to-treat cohort (N=10). Please add and include BMI metrics, as this study recruited an overweight sample.
Compliance with the weekly phone calls was quite low. Was there any association between phone call compliance and TRF adherence? Granted numbers are small but this is important for assessing your intervention design and making alterations for future studies.
Line 164. Change “decreased” to “occurred earlier” and “approximately a half hour” to “16 minutes”. Half hour change here is mis-representing the actual change in the average.
Line 177. Cite the survey question(s) you are referring to for this statement and the values that support for “most”. It was 4 of 7 that said fasting got easier. It was 7 of 9 that said they disagreed at all that eating over 8 hours was difficult.
Section beginning on Line 189. Lots of interesting findings here that are so important to publish! Particularly the low level of compliance/adherance due to mis-understanding. This factor must be considered in the Discussion and Abstract with respect to interpretation of findings AND mentioned quantitatively (3 with full protocol compliance to calls and TRF design) in the Abstract. These factors surely impacted the variables in Table 1. This should be explored. Please state how many participants were calorie restricting (Line 200).
Discussion.
Line 233. This statement is overly strong and should be tempered down. Only 9 completed and many were not compliant, i.e., ate calorie-containing foods during fasting times, ate only 1 meal per day, reduced their calorie intake, ate small snacks or low calorie foods. The data do not strong support answers regarding feasibility (which includes participants’ comprehension of the intervention design), tolerability (Were these scores higher because of the calorie intake during “fasting” times? Did that make them hungrier, moodier?), or safety (Did calorie intake during “fasting” times make it safer?). This study’s findings are important and interesting BUT the authors need to be transparent and measured in their interpretations of the findings given the very small population size and poor study compliance.
Line 236. Side effects and adverse outcomes should be reported in the Results section, not the Discussion section. Please add a full description of each of these to the end of the Results (add a Table, if that is warranted by number of events/issues reported).
Line 242. First time where it is clear that these are same participants as in ref #19. This sentence about likely reduced food intake needs to be edited/dropped. Many mouse studies and Sutton, et al. 2018 (Cell Metabolism) in middle-aged men have shown that TRF is associated with fat mass loss without calorie restriction.
You have a very small sample number. Throughout the entire Discussion, you must cite the numbers rather than use vague terms. For example, Line 245. Change “many” to or add number reporting this out of the number reporting. Line 246. Change “vast majority” to or add number reporting this out of the number reporting. Change “many” “most” etc. as specified above. Often, these are referring to only 4 people, who may or may not have been following the protocol.
Line 282. Limitations – cite how many got all the calls during intervention. Tiny sample size is a huge limitation (not mentioned at all). Comment on the effect of this on your power to make conclusive statements. Study design (pre-post) is another huge limitation (not mentioned). These limitations are expected for pilot studies but you must address them here. Discuss what next steps with respect to trial design – RCT, sufficiently-powered sample size, etc.
MINOR CONCERNS.
Line 20. Please rephrase. Seems as if participants were only offered beverage item options during feeding and fasting periods.
Reference needed for sentence ending on Line 78. This length of time will, of course, vary on activity levels, BMI, age, etc. This sentence is also not clear as to what types of studies this evidence comes from. Please clarify. A reader my surmise that is comes from TRF/IF studies.
Lines 79-80. Please site the original research study papers supporting this statement, not review by this manuscript’s senior author.
Line 90. Change to “single arm pre-post design.”
Line 85. Change “sustainability” to “feasibility”. This study doesn’t not address sustainability (long-term adherence) in a significant way.
Line 177. Describe specifically how this participant’s experience/mis-understanding was reported. Data above about end-of-study survey. This was collected differently and reader may be confused.
TABLES & FIGURES.
Figure 1. Nice to see individual eating start and stop patterns. What is the “N” for each timepoint. Please add. Not all were reached during the intervention period. Please add an additional data panel that shows each individual’s fasting intervals for each timepoint. Important to show baseline fasting times and how the lengths of those change during intervention. Very hard to guesstimate that in current figure state.
Table 1 and Figures 2-4. Why are the questions about IF and not TRF? Seems that might have been confusing to the participants. I don’t think that Figures 2-4 are necessary. They don’t show anything different than what is shown in Table 1.`
Author Response
Reviewer 1
Open Review
(x) I would not like to sign my review report
( ) I would like to sign my review report
English language and style
( ) Extensive editing of English language and style required
( ) Moderate English changes required
(x) English language and style are fine/minor spell check required
( ) I don't feel qualified to judge about the English language and style
|
Yes |
Can be improved |
Must be improved |
Not applicable |
|
|
Does the introduction provide sufficient background and include all relevant references? |
(x) |
( ) |
( ) |
( ) |
|
Is the research design appropriate? |
(x) |
( ) |
( ) |
( ) |
|
Are the methods adequately described? |
(x) |
( ) |
( ) |
( ) |
|
Are the results clearly presented? |
( ) |
(x) |
( ) |
( ) |
|
Are the conclusions supported by the results? |
( ) |
( ) |
(x) |
( ) |
Comments and Suggestions for Authors
I am happy to see this pilot pre-post TRF study done in an older population as a first step in evaluating the feasibility, tolerability and safety of the intervention regimen in this population. The authors are to be congratulated on their consideration of these factors and population. The findings are promising. Some concerns dampen my enthusiasm of the manuscript in its present form.
MAJOR CONCERNS.
Abstract.
Line 12. Slight mis-characterization of Delete “few or”. TRF design is that no calories are to be consumed during the fasting period. Further, TRF is not a prescribed fasting length of 12-14 hrs, there are shorter and longer regimens that have been published. Best to describe simply that 12-14 hrs is “often” or “commonly” used in study designs. Here in abstract, also Line 74.
Response: We have deleted the words “few or” and added the word “commonly” as suggested.
Study design should be specifically mentioned in abstract: Pilot, one-arm pre-post intervention study.
Response: We would like to thank the review for this note and have added the wording “one-arm pre-post intervention study” to the abstract as suggested.
Line 26. Demographic data should be for intention-to-great cohort (N=10). Unclear, whether data show is for the 10 enrolled or the 9 who completed. Please address and add BMI metrics.
Response: We thank the reviewer for this comment and have now modified this sentence in the abstract to make it clear that the demographic data refers to the intention to treat cohort. Specifically, we now state:
“Of the 10 participants who commenced the study (mean age = 77.1 y; 6 women, 4 men), nine completed the entire protocol.”
How and how often data about feasibility, acceptability, and other input on intervention outcomes was collected should be reported in abstract, as these are main outcomes.
Response: We fully agree and apologize for this omission. We have now inserted the following sentence in the Methods section of the abstract.
“At the end of the intervention, participants completed an exit interview and a study-specific Diet Satisfaction Survey (Table 1) to assess their satisfaction, feasibility, and overall experience with the study intervention.”
Outcomes should be quantitatively (not using percentiles) in abstract. Terms like “majority”, “some”, and “rare” are too vague.
Response: We thank the reviewer for this important note. We have removed the vague language and replaced it with more direct/explicit wording to better describe our results.
The “Methods” section of the abstract should contain the intervention length and the design (daily 16hr fast).
Response: We thank the reviewer for this catch. We have added the intervention length and the design as suggested. Specifically, we now state:
“Participants were instructed to fast for approximately 16 h per day with the daily target range between 14–18 h.”
Challenges in the participants’ understanding of the protocol is an important finding. Strongly suggest adding here, or it will be missed by many who would find it very useful to know.
Response: We thank the reviewer for this suggestion and have added additional information so it is not overlooked by the reader. Specifically, we now state:
“Six participants, however, did not fully understand the requirements of the fasting regimen, despite being provided with specific instructions and a pictorial guide at a baseline visit. This suggests that more instruction and/or participant contact is needed in the early stages of a TRF intervention to promote adherence.”
Methods.
Line 87. Please clarify. Is this manuscript a report of secondary outcomes analyses from the same research study presented in reference #19? If so, this must be made completely clear that this the same study and participants and not simply the same protocol. It is the same study, are there any data presented here (text, figures, tables) that were presented in ref #19?
Response: We thank the reviewer for this important note. This section has been reworded to clearly show this is secondary data from the same study and participants as were involved in reference #25 (previously ref #19).
Line 87. Define “mild to moderate functional limitations” in this manuscript. Where were the participants living? Please provide information on this. Free-living or living in senior centers with common dining? These factors are important for this population as well as the intervention feasibility.
Response: We thank the reviewer for these comments and questions. We have now clarified that these participants were all living independently in the community, and also noted that additional details about the participants have been described previously in reference #25 (previously ref #19).
Line 91. List of secondary outcomes sounds like ref #19. No reference provided here. Please add. Why were the “secondary outcomes” of the study reported first? Seems that this manuscript is reporting secondary outcomes of the overall study. Please explain.
Response: The reviewer is correct that the secondary outcomes were reported in ref #25 (previously reference #19), which we have now added to clarify that the original manuscript reported the primary and secondary outcomes as described in the Methods section. The current manuscript is reporting additional information on the real-world advantages, disadvantages, and challenges to adopting a TRF eating pattern among participants aged 65 and over.
The methods section should include a CONSORT-type diagram. There is a pilot study version of this. What was recruitment efficacy like, how many individuals screened prior to enrollment, why were people excluded and screening out, etc.?
Response: We thank the reviewer for this suggestion. A CONSORT-type diagram for this pilot study was included in the previous publication (reference #25).
For the reviewer’s convenience, we have pasted this Consort Diagram below.
Results.
Line 151. It is standard to report demographics of the intention-to-treat cohort (N=10). Please add and include BMI metrics, as this study recruited an overweight sample.
Response: We agree and have reported demographics and BMI metrics previously in reference #25.
Compliance with the weekly phone calls was quite low. Was there any association between phone call compliance and TRF adherence? Granted numbers are small but this is important for assessing your intervention design and making alterations for future studies.
Response: There was only a moderate positive correlation between weekly phone call completion with adherence levels (.024) during the study. This may be because of the restricted range of adherence in that participants were generally compliant (range = 64% – 100%) with only two participants reporting adherence levels below 75%. This correlation may also be lower than expected as the one participant who reported 100% adherence only completed one phone call. Taken together, this suggests that the phone calls may only have a moderate effect on adherence and that future studies should consider evaluating the effects of in person and/or online video coaching as compared to phone calls.
Line 164. Change “decreased” to “occurred earlier” and “approximately a half hour” to “16 minutes”. Half hour change here is mis-representing the actual change in the average.
Response: We thank the reviewer for these suggestions. We have changed the wording and clearly state the decrease in eating stop time over the 4 weeks was 27 minutes, rather than ‘approximately a half hour.’
Line 177. Cite the survey question(s) you are referring to for this statement and the values that support for “most”. It was 4 of 7 that said fasting got easier. It was 7 of 9 that said they disagreed at all that eating over 8 hours was difficult.
Response: We have changed the vague terms to clearly state number of participants and reference figure 2 to coincide with this, which lists the survey questions.
Section beginning on Line 189. Lots of interesting findings here that are so important to publish! Particularly the low level of compliance/adherence due to mis-understanding. This factor must be considered in the Discussion and Abstract with respect to interpretation of findings AND mentioned quantitatively (3 with full protocol compliance to calls and TRF design) in the Abstract. These factors surely impacted the variables in Table 1. This should be explored. Please state how many participants were calorie restricting (Line 200).
Response: As suggested, we have now mentioned the role that participant’s poor comprehension of the TRF intervention may have had in affecting their levels of adherence in the Abstract and first paragraph of the Discussion. We have also removed the vague terms have been removed to state exact participant numbers in line 200.
Specifically, we state in the Abstract – “The findings of the current trial suggest that TRF is an eating approach that is well tolerated by most older adults. Six participants, however, did not fully understand the requirements of the fasting regimen, despite being provided with specific instructions and a pictorial guide at a baseline visit. This suggests that more instruction and/or participant contact is needed in the early stages of a TRF intervention to promote adherence.”
And also state the following in the first paragraph of the Discussion - “The findings of the current trial suggest that TRF is an eating approach that is well tolerated by most older adults. Six participants, however, did not fully understand the requirements of the fasting regimen, despite being provided with specific instructions and a pictorial guide at a baseline visit. Among these six participants, three reported consuming snacks during fasting periods, two participants confused low-calorie with no-calorie items, and one participant thought they were only allowed to eat one meal a day. Thus, these findings suggest that more instruction and/or participant contact is needed in the early stages of a TRF intervention to promote adherence.”
Discussion.
Line 233. This statement is overly strong and should be tempered down. Only 9 completed and many were not compliant, i.e., ate calorie-containing foods during fasting times, ate only 1 meal per day, reduced their calorie intake, ate small snacks or low calorie foods. The data do not strong support answers regarding feasibility (which includes participants’ comprehension of the intervention design), tolerability (Were these scores higher because of the calorie intake during “fasting” times? Did that make them hungrier, moodier?), or safety (Did calorie intake during “fasting” times make it safer?). This study’s findings are important and interesting BUT the authors need to be transparent and measured in their interpretations of the findings given the very small population size and poor study compliance.
Response: We agree and have now revised the statement to be less definitive and also highlight the issues related to misunderstanding the protocol in the first paragraph of the discussion. Specifically, we now state that:
“The findings of the current trial suggest that TRF is an eating approach that is well tolerated by most older adults. Six participants, however, did not fully understand the requirements of the fasting regimen, despite being provided with specific instructions and a pictorial guide at a baseline visit. Among these six participants, three reported consuming snacks during fasting periods, two participants confused low-calorie with no-calorie items, and one participant thought they were only allowed to eat one meal a day. Thus, these findings suggest that more instruction and/or participant contact is needed in the early stages of a TRF intervention to promote adherence.”
Line 236. Side effects and adverse outcomes should be reported in the Results section, not the Discussion section. Please add a full description of each of these to the end of the Results (add a Table, if that is warranted by number of events/issues reported).
Response: Side effects and adverse outcomes from this pilot study have been previously reported in detail in reference #25 (previously #19). As suggested, we have now added the following sentences to the Results section:
“Few adverse events were reported during this intervention. Specifically, two participants experienced headaches during fasting periods, which resolved following an increase in water intake. One participant experienced dizziness, which resolved after having a small snack.”
Line 242. First time where it is clear that these are same participants as in ref #19. This sentence about likely reduced food intake needs to be edited/dropped. Many mouse studies and Sutton, et al. 2018 (Cell Metabolism) in middle-aged men have shown that TRF is associated with fat mass loss without calorie restriction.
Response: As suggested, we have now removed the sentence stating that participants likely reduced food intake.
You have a very small sample number. Throughout the entire Discussion, you must cite the numbers rather than use vague terms. For example, Line 245. Change “many” to or add number reporting this out of the number reporting. Line 246. Change “vast majority” to or add number reporting this out of the number reporting. Change “many” “most” etc. as specified above. Often, these are referring to only 4 people, who may or may not have been following the protocol.
Response: As suggested, we have removed vague terms and replaced these terms with the
actual number of participants who fit in each category.
Line 282. Limitations – cite how many got all the calls during intervention. Tiny sample size is a huge limitation (not mentioned at all). Comment on the effect of this on your power to make conclusive statements. Study design (pre-post) is another huge limitation (not mentioned). These limitations are expected for pilot studies but you must address them here. Discuss what next steps with respect to trial design – RCT, sufficiently-powered sample size, etc.
Response: We thank the reviewer for these comments. We fully agree with the limitations described above and previously listed these limitations in reference #25 (previously #19). We have also now added these limitations to the current manuscript, as well as described next steps in this line of research. Specifically, we now state
“Future studies should utilize larger sample sizes to ensure sufficient power to detect both pre-post differences and between group differences.”
MINOR CONCERNS.
Line 20. Please rephrase. Seems as if participants were only offered beverage item options during feeding and fasting periods.
Response: This has been rephrased to note that participants were offered both food and beverage items that were allowable during feeding vs. fasting periods.
Reference needed for sentence ending on Line 78. This length of time will, of course, vary on activity levels, BMI, age, etc. This sentence is also not clear as to what types of studies this evidence comes from. Please clarify. A reader my surmise that is comes from TRF/IF studies.
Response: We have now added references to this sentence to clarify the source of this information, which were intermittent fasting studies but not TRF studies.
Lines 79-80. Please site the original research study papers supporting this statement, not review by this manuscript’s senior author.
Response: As suggested, we have now inserted the original study papers supporting this statement.
Line 90. Change to “single arm pre-post design.”
Response: The wording has now been updated.
Line 85. Change “sustainability” to “feasibility”. This study doesn’t not address sustainability (long-term adherence) in a significant way.
Response: We appreciate the reviewer’s recommendation and have now changed “sustainability” to “feasibility.”
Line 177. Describe specifically how this participant’s experience/mis-understanding was reported. Data above about end-of-study survey. This was collected differently and reader may be confused.
Response: Thank you for this comment. This data was collected at the end-of-study survey. We have clarified this sentence to state
“One participant reported uncomfortable hunger during the study, however, this participant misunderstood the protocol and was only eating one meal a day.”
TABLES & FIGURES.
Figure 1. Nice to see individual eating start and stop patterns. What is the “N” for each timepoint. Please add. Not all were reached during the intervention period. Please add an additional data panel that shows each individual’s fasting intervals for each timepoint. Important to show baseline fasting times and how the lengths of those change during intervention. Very hard to guesstimate that in current figure state.
Response: Figure 1 has been updated to show group means for start and stop times for each of the four weeks. We have also added the following figure caption to clarify that each line represents a participant and that the total number of participants displayed was 9 participants. Specifically, we now state the following:
“Figure 1. Self-selected start and stop times for each participant’s eating window. N=9 for each of the four weeks. Each participant’s average weekly self-reported start/stop times are indicated by differing colors, with each line representing a single participant. The time between ‘Start Time’ and ‘Stop Time’ is indicative of each participant’s eating window.”
Table 1 and Figures 2-4. Why are the questions about IF and not TRF? Seems that might have been confusing to the participants. I don’t think that Figures 2-4 are necessary. They don’t show anything different than what is shown in Table 1.
Response: We agree that the information provided in Table 1 encompasses the same information as was provided in Figures 2-4. If possible, we would like to include the figures as supplementary files as they may be helpful to some readers.
Reviewer 2 Report
Article Summary:
This study reports on adherence and diet satisfaction during a 4-week, single-arm, time-restricted feeding (TRF) intervention in 9 older adults. While some of the data and insights are interesting, there are several drawbacks to this study, including the small sample size (N=9) and small data set (only numerical data on adherence and a single survey are presented). Although an exit interview was performed and discussed in this manuscript, the exit interview data was not quantified, as it should have been. Furthermore, the manuscript lacks statistics, such as standard deviations to characterize adherence (e.g., lines 160-161). Another significant issue is that the survey used was custom-designed and unvalidated. It lacks some of the principles of good survey design and some of the authors’ interpretations of the survey questions are not supported by wording of the questions themselves. For instance, the authors state that the participants rated the intervention as “easy,” but the participants were only asked if the intervention became easier over time, which is a different question. Overall, the impact of this manuscript is limited given the very small sample size and lower quality of data.
Other Comments:
- Abstract: TRF interventions with fasting durations of longer than 16 hours have been studied, so the definition that the authors use for TRF is not quite correct. Also, the definition of TRF in the Abstract conflicts with the definition provided in the Introduction.
- Adverse events are reported in the Abstract but are missing from the Results section. All AEs reported should be listed and quantified as to frequency, etc.
- The Introduction focuses too much on sarcopenia and mitochondrial function (which are unrelated to the data presented) and too little on TRF. Paragraphs 2-4 are out-of-place and should be re-written.
- Since the topic of the manuscript is TRF and behavioral factors, the Introduction should reflect this. As is, the Introduction does not sufficiently review the literature on (1) TRF studies in humans and (2) diet satisfaction and adherence to intermittent fasting interventions. Similarly, comparisons are not made to the existing literature in the Discussion section.
- Adherence was defined as a self-reported eating time within a 6-10-hour window of choice, but criteria on cutoffs for what was considered adherent was not provided (e.g., met criteria on all days or met a specific threshold). Did participants need to complete this on every single day of the study to be considered adherent? Generally, adherence is expressed as a percentage of days or another non-binary metric.
- There is insufficient detail on the Exit Interview in the Methods section. The manuscript would benefit from presentation of the phone counseling and Exit Interview questions.
- Figure 1: Please also display the group means.
- Some of the custom-designed survey questions are not well-designed or well-worded, from a survey design perspective. The survey questions and possible responses could be improved by (a) counterbalancing positive and negative study attributes, (b) aligning answer options with the questions, and (c) less ambiguous wording.
- Sections 3.2 and 3.3 do not discuss the results of the survey in sufficient detail, relative to Section 3.1, which is more detailed.
- The data are presented in a repetitious way: Figures 2-4 are just visual representations of the data presented in the table and should be deleted.
- The references are incorrectly formatted.
- The authors indicate that data on the foods that participants ate was collected. Given that data exists, why is the behavioral data on food intake not presented here?
- A second coder should have been used to cross-check the coding scheme.
Author Response
Reviewer 2
Open Review
(x) I would not like to sign my review report
( ) I would like to sign my review report
English language and style
( ) Extensive editing of English language and style required
( ) Moderate English changes required
(x) English language and style are fine/minor spell check required
( ) I don't feel qualified to judge about the English language and style
|
Yes |
Can be improved |
Must be improved |
Not applicable |
|
|
Does the introduction provide sufficient background and include all relevant references? |
( ) |
( ) |
(x) |
( ) |
|
Is the research design appropriate? |
( ) |
(x) |
( ) |
( ) |
|
Are the methods adequately described? |
( ) |
(x) |
( ) |
( ) |
|
Are the results clearly presented? |
( ) |
(x) |
( ) |
( ) |
|
Are the conclusions supported by the results? |
( ) |
(x) |
( ) |
( ) |
Comments and Suggestions for Authors
Article Summary:
This study reports on adherence and diet satisfaction during a 4-week, single-arm, time-restricted feeding (TRF) intervention in 9 older adults. While some of the data and insights are interesting, there are several drawbacks to this study, including the small sample size (N=9) and small data set (only numerical data on adherence and a single survey are presented). Although an exit interview was performed and discussed in this manuscript, the exit interview data was not quantified, as it should have been. Furthermore, the manuscript lacks statistics, such as standard deviations to characterize adherence (e.g., lines 160-161). Another significant issue is that the survey used was custom-designed and unvalidated. It lacks some of the principles of good survey design and some of the authors’ interpretations of the survey questions are not supported by wording of the questions themselves. For instance, the authors state that the participants rated the intervention as “easy,” but the participants were only asked if the intervention became easier over time, which is a different question. Overall, the impact of this manuscript is limited given the very small sample size and lower quality of data.
Other Comments:
- Abstract: TRF interventions with fasting durations of longer than 16 hours have been studied, so the definition that the authors use for TRF is not quite correct. Also, the definition of TRF in the Abstract conflicts with the definition provided in the Introduction.
- Response: We thank the reviewer for catching this inconsistency. We have changed the numbering in the abstract to coincide with the definition within the introduction. Additionally, we have added wording to clarify that this time period is commonly used for TRF protocol.
- Adverse events are reported in the Abstract but are missing from the Results section. All AEs reported should be listed and quantified as to frequency, etc.Specifically, we have added the following statement to the Results section:
- “Few adverse events were reported during this intervention. Specifically, two participants experienced headaches during fasting periods, which resolved following an increase in water intake. One participant experienced dizziness, which resolved after having a small snack.”
- Response: As suggested, we have reported the AEs in the Results section as well as the Abstract.
- The Introduction focuses too much on sarcopenia and mitochondrial function (which are unrelated to the data presented) and too little on TRF. Paragraphs 2-4 are out-of-place and should be re-written.
- Response: The reason that the manuscript focused on sarcopenia and mitochondrial function was because of the theme of this Special Issue. We do agree with the reviewer, however, that some information on sarcopenia and mitochondrial function could be removed. Accordingly, we have removed the paragraph below from the Introduction.
“Given these links in later life, a logical next question is why mitochondrial function decreases with aging. Although this remains a topic of debate, a growing body of literature indicates that there is a clear connection between mitochondrial biogenesis, metabolic function and fuel metabolism.[10, 11] Specifically, efficient skeletal muscle bioenergetics appears to be strongly linked to the metabolic flexibility of the mitochondria.[12, 13] This suggests muscle health maintenance in aging relies on the ability of the mitochondria to shift from using glucose from glycogenolysis to fatty acids and fatty acid-derived ketones as a source of energy. Thus, therapeutic interventions that shift the source of energy utilized by the mitochondria should lead to improvements in cellular health, muscle quality and physical function in older adults.”
- Since the topic of the manuscript is TRF and behavioral factors, the Introduction should reflect this. As is, the Introduction does not sufficiently review the literature on (1) TRF studies in humans and (2) diet satisfaction and adherence to intermittent fasting interventions. Similarly, comparisons are not made to the existing literature in the Discussion section. “In contrast to traditional caloric restriction paradigms, food is not consumed during designated fasting time periods but is typically not restricted during designated eating time periods. The length of the fasting time period can also vary but is frequently 12 or more continuous hours. There are many types of intermittent fasting approaches but the two most popular and well-studied approaches are alternate day fasting (ADF) or alternate day modified fasting (ADMF) and TRF. Alternate day or alternate day modified fasting, involves consuming no or very little food on fasting days and then alternating with a day of unrestricted food intake or a “feast” day. Time restricted feeding interventions differ from ADF interventions in that individuals engage in daily fasts between 14-18 hours. Findings from a recent review indicate participants generally have high levels of adherence (range = 77% to 98%) with no serious adverse events to fasting regimens ranging in duration from two weeks to one year.”21
- Response: We agree with the reviewer that additional information on intermittent fasting studies and TRF in particular in humans would be helpful to the reader. Accordingly, we have now inserted the paragraph below in the introduction.
We have also added the following paragraph to the discussion.
Despite strong evidence indicating that lifestyle intervention programs involving diet, exercise, and behavior modification can reduce risk factors for many chronic diseases and improve physical function, long-term adherence to lifestyle interventions to date is notoriously low.[34] Consequently, the “adherence problem” represents an important challenge to weight loss interventions.[34] Findings from a recent review of 27 studies indicate that participants generally have high levels of adherence (range = 77% to 98%) to different types of fasting regimens, including ADF and TRF.[21] Thus, future trials are needed to evaluate the potential that this eating pattern may have for enhancing long-term weight loss.
- Adherence was defined as a self-reported eating time within a 6-10-hour window of choice, but criteria on cutoffs for what was considered adherent was not provided (e.g., met criteria on all days or met a specific threshold). Did participants need to complete this on every single day of the study to be considered adherent? Generally, adherence is expressed as a percentage of days or another non-binary metric.
- Response: Thank you for this comment. Under section 2.2 Adherence, we have explained that participants were considered adherent if they reported fasting between 14-18 hours per day during the intervention. In the Results section we have added the following, which has been previously published: “Self-reported mean adherence to the TRF regimen was 84%, measured by daily eating time logs.”
- There is insufficient detail on the Exit Interview in the Methods section. The manuscript would benefit from presentation of the phone counseling and Exit Interview questions.
- Response: We agree with this recommendation and would like to include the questions from these forms in the supplemental material.
- Figure 1: Please also display the group means.
- Response: We thank the reviewer for this suggested and have added weekly group means to Figure 1.
- Some of the custom-designed survey questions are not well-designed or well-worded, from a survey design perspective. The survey questions and possible responses could be improved by (a) counterbalancing positive and negative study attributes, (b) aligning answer options with the questions, and (c) less ambiguous wording.
- Response: We thank the reviewer for this comment and will do our best to improve this instrument in future trials.
- Sections 3.2 and 3.3 do not discuss the results of the survey in sufficient detail, relative to Section 3.1, which is more detailed.
- Response: We have now revised Sections 3.2 and 3.3 to match Section 3.1 in the level of detail provided within each section.
- The data are presented in a repetitious way: Figures 2-4 are just visual representations of the data presented in the table and should be deleted.
- Response: We agree that the information provided in Table 1 encompasses the same information as was provided in Figures 2-4. If possible, we would like to include the figures as supplementary files as they may be helpful to some readers.
- The references are incorrectly formatted.
- Response: We thank the reviewer for catching this error and have now correctly formatted the references.
- The authors indicate that data on the foods that participants ate was collected. Given that data exists, why is the behavioral data on food intake not presented here?Specifically, in the abstract, we state “Participants were instructed to complete daily eating time logs by recording the times at which they first consumed calories and when they stopped consuming calories.” Response: We agree and have noted this as a limitation of the present study. In the future, a second coder will be used to cross-check the coding scheme. For this reason, the present manuscript focused on the responses to the Diet Satisfaction Survey in the Results and Discussion section.
- A second coder should have been used to cross-check the coding scheme.
- Response: The data on the actual foods that participants ate was not collected. Rather, participants reported only the times in which they start and stopped each day. For this reason, we have referred to the diary participants completed as an “eating time log” rather than a “food diary.”
Reviewer 3 Report
The manuscript by Lee SA et al trialed the time restricted feeding in 10 older adults. The authors reported that 9 of the 10 participants completed the entire protocol. It seems not too difficult for the participants to adapt a 16-hour fast protocol. Increased energy was reported by some participants, with greater self-reported activity levels in yardwork and light exercise. Adverse events were rare. The study is interesting, and the data is encouraging. I just have one suggestion for the manuscript:
In the introduction, the authors spent quite some words to discuss mitochondria and muscle health, as well as exercise and mitochondria. However, there is no mitochondrial data in the manuscript. I would suggest the authors change the introduction a bit, to make it more relevant to the current study.
Author Response
Open Review
( ) I would not like to sign my review report
(x) I would like to sign my review report
English language and style
( ) Extensive editing of English language and style required
( ) Moderate English changes required
(x) English language and style are fine/minor spell check required
( ) I don't feel qualified to judge about the English language and style
|
Yes |
Can be improved |
Must be improved |
Not applicable |
|
|
Does the introduction provide sufficient background and include all relevant references? |
( ) |
(x) |
( ) |
( ) |
|
Is the research design appropriate? |
(x) |
( ) |
( ) |
( ) |
|
Are the methods adequately described? |
(x) |
( ) |
( ) |
( ) |
|
Are the results clearly presented? |
(x) |
( ) |
( ) |
( ) |
|
Are the conclusions supported by the results? |
(x) |
( ) |
( ) |
( ) |
Comments and Suggestions for Authors
The manuscript by Lee SA et al trialed the time restricted feeding in 10 older adults. The authors reported that 9 of the 10 participants completed the entire protocol. It seems not too difficult for the participants to adapt a 16-hour fast protocol. Increased energy was reported by some participants, with greater self-reported activity levels in yardwork and light exercise. Adverse events were rare. The study is interesting, and the data is encouraging. I just have one suggestion for the manuscript:
In the introduction, the authors spent quite some words to discuss mitochondria and muscle health, as well as exercise and mitochondria. However, there is no mitochondrial data in the manuscript. I would suggest the authors change the introduction a bit, to make it more relevant to the current study.
Response: The reason that the manuscript focused on sarcopenia and mitochondrial function was because of the theme of this Special Issue. We do agree with the reviewer, however, that some information on sarcopenia and mitochondrial function could be removed. Accordingly, we have removed the paragraph below from the Introduction.
“Given these links in later life, a logical next question is why mitochondrial function decreases with aging. Although this remains a topic of debate, a growing body of literature indicates that there is a clear connection between mitochondrial biogenesis, metabolic function and fuel metabolism.[10, 11] Specifically, efficient skeletal muscle bioenergetics appears to be strongly linked to the metabolic flexibility of the mitochondria.[12, 13] This suggests muscle health maintenance in aging relies on the ability of the mitochondria to shift from using glucose from glycogenolysis to fatty acids and fatty acid-derived ketones as a source of energy. Thus, therapeutic interventions that shift the source of energy utilized by the mitochondria should lead to improvements in cellular health, muscle quality and physical function in older adults.”
To make the introduction more relevant to the current study, we have also added the following paragraph to the Introduction.
In contrast to traditional caloric restriction paradigms, food is not consumed during designated fasting time periods but is typically not restricted during designated eating time periods. The length of the fasting time period can also vary but is frequently 12 or more continuous hours. There are many types of intermittent fasting approaches but the two most popular and well-studied approaches are alternate day fasting (ADF) or alternate day modified fasting (ADMF) and TRF. Alternate day or alternate day modified fasting, involves consuming no or very little food on fasting days and then alternating with a day of unrestricted food intake or a “feast” day. Time restricted feeding interventions differ from ADF interventions in that individuals engage in daily fasts between 14-18 hours. Findings from a recent review indicate participants generally have high levels of adherence (range = 77% to 98%) with no serious adverse events to fasting regimens ranging in duration from two weeks to one year. [17]